# Adenoma to Colorectal Cancer Estimated Transition Rates Stratified by BMI Categories—A Cross-Sectional Analysis of Asymptomatic Individuals from Screening Colonoscopy Program

**DOI:** 10.3390/cancers14010062

**Published:** 2021-12-23

**Authors:** Piotr Spychalski, Jarek Kobiela, Paulina Wieszczy, Marek Bugajski, Jaroslaw Reguła, Michał F. Kaminski

**Affiliations:** 1Department of General, Endocrine and Transplant Surgery, Medical University of Gdansk, 80-210 Gdansk, Poland; jaroslaw.kobiela@gumed.edu.pl; 2Centre of Postgraduate Medical Education, Department of Gastroenterology, Hepatology and Clinical Oncology, 01-813 Warsaw, Poland; p.wieszczy@gmail.com (P.W.); marek.bugajski@pib-nio.pl (M.B.); jregula@coi.waw.pl (J.R.); mfkaminski@pib-nio.pl (M.F.K.); 3Clinical Effectiveness Research Group, Institute of Health and Society, Faculty of Medicine, University of Oslo, NO-0316 Oslo, Norway; 4Department of Oncological Gastroenterology, Maria Sklodowska-Curie National Research Institute of Oncology, 01-813 Warsaw, Poland; 5Department of Cancer Prevention, Maria Sklodowska-Curie National Research Institute of Oncology, 01-813 Warsaw, Poland

**Keywords:** colorectal cancer, colorectal cancer prevalence, colorectal adenoma, screening, colonoscopy, colorectal cancer epidemiology

## Abstract

**Simple Summary:**

Most colorectal cancers assumedly develop from precursor lesions, i.e., colorectal adenomas (adenoma-carcinoma sequence). Epidemiological and clinical data supporting this hypothesis are limited. Therefore, the aim of the present study was to estimate relative dynamics of colorectal adenoma-carcinoma sequence for groups of screenees stratified by body mass. We have analyzed 163,129 individuals that underwent screening colonoscopy and calculated adjusted prevalences of colorectal adenomas and colorectal. Based on that data we have found that obese individuals are more likely to develop adenomas, advanced adenomas and early CRC but less likely to progress to advanced CRC. Therefore, this study provides new evidence that obesity paradox exists for colorectal cancer.

**Abstract:**

Most colorectal cancers (CRC) assumedly develop from precursor lesions, i.e., colorectal adenomas (adenoma-carcinoma sequence). Epidemiological and clinical data supporting this hypothesis are limited. Therefore, the aim of the present study is to estimate relative dynamics of colorectal adenoma-carcinoma sequence for groups of screenees stratified by BMI (body mass index) based on prevalence data from Polish Colonoscopy Screening Program (PCSP). We performed a cross-sectional analysis of database records of individuals who entered the national opportunistic colonoscopy screening program for CRC in Poland. We calculated prevalence of adenomas and CRCs adjusted for sex, 5-year age group, family history of CRC, smoking, diabetes and use of aspirin, hormonal therapy and proton-pump inhibitors use. Thereafter we calculated estimated transition rate (eTR) with confidence intervals (CIs) defined as adjusted prevalence of more advanced lesion divided by adjusted prevalence of less advanced lesion. All analyzes were stratified according to the BMI categories: normal (BMI 18.0 to <25.0), overweight (BMI 25.0 to <30.0) and obese (BMI ≥ 30.0). Results are reported in the same respective order. After exclusions we performed analyses on 147,385 individuals. We found that prevalence of non-advanced adenomas is increasing with BMI category (12.19%, 13.81%, 14.70%, respectively; *p* < 0.001). Prevalence of advanced adenomas was increasing with BMI category (5.20%, 5.77%, 6.61%, respectively; *p* < 0.001). Early CRCs prevalence was the highest for obese individuals (0.55%) and the lowest for overweight individuals (0.44%) with borderline significance (*p* = 0.055). For advanced CRC we found that prevalence seems to be inversely related to BMI category, however no statistically significant differences were observed (0.35%, 0.31%, 0.28%; *p* = 0.274). eTR for non-advanced adenoma to advanced adenoma is higher for obese individuals than for overweight individuals with bordering CIs (42.65% vs. 41.81% vs. 44.95%) eTR for advanced adenoma to early CRC is highest for normal individuals, however CIs are overlapping with remaining BMI categories (9.02% vs. 7.67% vs. 8.39%). eTR for early CRC to advanced CRC is lower for obese individuals in comparison to both normal and overweight individuals with marginally overlapping CIs (73.73% vs. 69.90% vs. 50.54%). Obese individuals are more likely to develop adenomas, advanced adenomas and early CRC but less likely to progress to advanced CRC. Therefore, this study provides new evidence that obesity paradox exists for colorectal cancer.

## 1. Introduction

An acclaimed paradigm states that most CRCs develop from precursor lesions, i.e., colorectal adenomas (CRA) [1]. This phenomenon is called adenoma-carcinoma sequence. The model assumes that natural history of CRC has the following steps: normal epithelium of colon and rectum transforms into colorectal adenoma. Later, adenomas transform into advanced adenomas. After crucial mutations appear, advanced adenoma transforms into early colorectal cancer, which, if not detected and treated, progresses trough stages, defined by American Joint Committee on Cancer (AJCC) [2].

Epidemiological and clinical data supporting this hypothesis are limited due to ethical considerations. Only one observational study on living individuals has been conducted—before the era of colonoscopy [3]. Consequently, researchers attempted to assess epidemiology of abovementioned lesions with use of post mortem examinations [4,5,6,7]. It is known, that not all CRAs will progress to CRC. It is crucial to assess the probability of that progression as well as to identify lesions that are at the greatest risk of progression and estimate the time of that progression. To date this has been only partially accomplished, to some extent due to study design difficulties. Screening colonoscopy programs, such as Polish Colonoscopy Screening Program, open new possibilities of observing this phenomenon [8], including a cross-sectional design. Such design enables assessment of prevalence of lesions found in the screened population, what can be used to observe the dynamic of cancer development. Facilitating such methodology to analyze adenoma-carcinoma sequence in CRC may benefit the current state of knowledge, especially given the substantial numbers of individuals undergoing screening.

Our team has recently reported that distribution of stages of CRC diagnosed in a screening setting differs among groups of individuals stratified by BMI [9]. In that study we concluded that obese individuals have higher proportion of early CRC, when compared to non-obese. We hypothesized that this phenomenon could be potentially explained by an increased rate of adenoma formation without a proportional increase in CRC progression rate, or with a decrease in progression rate. Therefore, the aim of the present study is to estimate relative dynamics of colorectal adenoma-carcinoma sequence for groups of screenees stratified by BMI based on prevalence data on asymptomatic individuals screened within PCSP.

## 2. Materials and Methods

### 2.1. Study Design

We performed a cross-sectional analysis of database records of individuals who participated in the national opportunistic colonoscopy screening program for CRC in Poland, from January 2007 through December 2011. The database (accessed on 20 August 2018) contains demographic data, colonoscopy results, self-reported data on weight and height, and colorectal findings from 114 screening centers throughout Poland.

The research proposal was reviewed by the Bioethical Committee at the Maria Sklodowska-Curie Memorial Cancer Centre and Institute of Oncology on 3th of October 2014 and was judged to be exempt from oversight, as data was deidentified and could not be linked to a specific subject. Study protocol conforms to the 1975 Declaration of Helsinki, as reflected by abovementioned exemption by Bioethical Committee. Written informed consent was obtained from all participants entering the National Colorectal Cancer Screening Program.

### 2.2. Screening Procedures

Asymptomatic participants between the ages of 50 years and 66 years (40 years and 66 years in case of positive family history of CRC) were offered screening. Exclusion criteria were clinical suspicion of CRC; criteria for Lynch syndrome, familial adenomatous polyposis, or inflammatory bowel disease; and colonoscopy within the preceding 10 years [8,10,11]. Before colonoscopy, all participants were asked to fill in an epidemiological questionnaire including data on self-reported weight and height. Screening colonoscopy procedures were previously described in detail [8,10,11].

### 2.3. Study Definitions

(a).Early CRC is defined as CRC in clinical stage 1 and clinical stage 2 according to AJCC. Advanced CRC is defined as CRC in clinical stage 3 and clinical stage 4 according to AJCC.(b).Advanced adenoma is defined as adenomatous lesion that:—has histologically proven high-grade dysplasia OR/AND—is ≥10 mm large OR/AND—has villous or tubulovillous component [12].(c).Non-advanced adenoma is defined as any adenomatous lesion that was not advanced adenoma or colorectal cancer.(d).BMI was calculated using weight [kg]/(height [m])^2^ and stratified according to the WHO classification [13]. First, second and third class of obesity were pooled together to achieve appropriate power of analysis. Underweight category was excluded from analysis due to low number of patients.

### 2.4. Statistical Methods

Data on crude prevalence of colorectal findings in every subgroup was abstracted from the database. This included: prevalence of non-advanced adenomas, prevalence of advanced adenomas, prevalence of CRC and prevalence of CRC stratified by AJCC stages (I, II, III, IV).

Thereafter prevalence of every lesion type was adjusted for: family history of CRC (1st degree vs. other or none) use of nicotine (never vs. current or former), history of diabetes (yes vs. no), use of aspirin (yes vs. no), use of proton pump inhibitors (yes vs. no), use of hormone replacement therapy in women (yes vs. no), gender and 5-years age group. 95% confidence intervals (CI) were calculated and chi-square test was used to compare the rates between the groups. Tests were performed at 0.05 significance level.

To assess dynamics of CRC progression following methods were used: (1) comparison of adjusted prevalence of specific colorectal findings between groups stratified by BMI; (2) comparison of transition rates (TR) of specific steps of adenoma-carcinoma sequence between groups stratified by BMI; (3) comparison of prevalence ratios in between BMI categories:
Adjusted prevalence of a colorectal lesion (non-advanced adenoma, advanced adenoma, CRC) is defined as number of individuals with a specific lesion divided by numbers of individuals screened multiplied by 100% and thereafter adjusted as specified above.Estimated transition rate (eTR) is defined as adjusted prevalence of more advanced lesion divided by adjusted prevalence of less advanced lesion (e.g., prevalence of CRC divided by prevalence of advanced adenomas etc.).P_NAA_—adjusted prevalence of non-advanced adenomasP_AA_—adjusted prevalence of advanced adenomasP_ECRC_—adjusted prevalence of early CRCP_ACRC_—adjusted prevalence of advanced CRCeTR_NAA__àAA_ = P_AA/_P_NAA_ × 100%eTR_AA__àECRC_ = P_ECRC/_P_AA_ × 100%eTR_ECRC__àACRC_ = P_ACRC/_P_ECRS_ × 100% et cetera.Prevalence ratio was calculated as follows: adjusted prevalence of BMI subgroup (overweight or obese) was divided by adjusted prevalence of normal BMI subgroup. This calculation was performed for every type of lesion (non-advanced adenoma, advanced adenoma, early CRC, advanced CRC).Calculations were carried out with Stata Statistical Software, v. 13.1 (Stata Corporation, College Station, TX, USA).

## 3. Results

A total of 163 129 individuals were drawn from the database. Subsequently 15,744 (9.7%) individuals were excluded due to missing data on history of nicotine use, ASA use, PPIs, and BMI (593, 6129, 1134, 7888, respectively). We performed analyses on remaining 147,385 screened asymptomatic individuals to assess to the adjusted prevalence of non-advanced adenomas, advanced adenomas, early CRC and advanced CRC. Baseline characteristics are presented in Table 1. Prevalence was adjusted for all factors presented in the baseline characteristics table.

### 3.1. Adjusted Prevalence

Results of calculations of crude and adjusted prevalence are presented in Table 2. Adjusted prevalence values calculated as described in Methods section are presented in Panels A–D of Figure 1. As shown, prevalence of non-advanced adenomas is increasing with BMI category (Panel A) with statistical significance. Similarly, prevalence of advanced adenomas is increasing with BMI category (Panel B) with statistical significance. For the analysis of early CRCs prevalence is the highest for obese individuals (0.55%) and the lowest for overweight individuals (0.44%) with borderline significance. Despite that BMI categories and prevalence of advanced CRC seem to be inversely related no statistically significant differences were observed (Panel D).

### 3.2. Transition Rates

eTR calculated as described in Methods section are presented in Figure 2. eTR for non-advanced adenoma to advanced adenoma is higher for obese individuals than for overweight individuals with borderline CIs. eTR for advanced adenoma to early CRC is highest for normal individuals, however CIs are overlapping with remaining BMI categories. eTR for early CRC do advanced CRC is lower for obese individuals in comparison to both normal and overweight individuals with marginally overlapping CIs. Table with eTRs with CIs is available in Appendix A.

### 3.3. Prevalence Ratios

Prevalence ratios for BMI categories are presented in Figure 3. 

## 4. Discussion

To our knowledge this is the first study to show that estimated dynamics of CRC progression throughout carcinogenesis steps differ between normal, overweight and obese individuals. On a substantial group of asymptomatic CRC screenees we show that overweight and obese individuals are more likely to be diagnosed with adenomas without increased risk to be diagnosed with advanced CRC, when compared to normal BMI individuals.

Adjusting prevalence for possible confounders enabled us to analyze results that were possibly the least biased by epidemiological characteristics of CRC patients. This is especially important for age and sex as it has been shown, that probability of transition from advanced adenoma to CRC increases with age gradient and differs between genders [14]. When analyzing prevalence per BMI category it can be found that obese individuals are at greater risk for developing colorectal adenomas. This is in accordance with literature data [15,16,17,18]. Early cancers from our sample show similar trend, with highest prevalence of CRC in obese individuals. For advanced cancers trend may have reversed—with highest prevalence in normal body weight individuals and lowest in obese individuals—however with no statistically significant difference. This provides further evidence that excessive body weight may be favorable in terms progression of CRC.

To better understand the dynamics of this observation we calculated transition rates (TRs) using adjusted prevalence. Similar methodology was already used by Brenner et al. [14]. However, in the abovementioned study authors used prevalence of advanced adenomas from a screening program and incidence of CRC drawn from national cancer database. This enabled calculation of annual risks of progression. In our study all variables were derived from screening program, with more coherent sample. We have aimed to calculate eTRs for early and advanced cancers, as well as adenomas and advanced adenomas. While Polish National Cancer Registry gathers information on CRC prevalence, incidence and stage of diagnosis it does not contain information on precursor lesions. Furthermore, Brenner et al. have adjusted TRs by a percentage of 85%—to account for adenomas that will never progress to CRC. This was based on previous assumptions by other researchers [19]. For our study similar approach did not seem necessary, as the primary goal was comparing dynamics between BMI subgroups instead of assessing accurate TRs. Therefore, estimated transition rates were calculated based on adjusted prevalence without correcting for cancers that are possibly not developing from conventional adenomas, as it may be with serrated polyps [20]. Based on the analysis of prevalence, we have shown that probability of transition from non-advanced adenoma to advanced adenoma is the greatest for obese individuals. However, this trend may fade or even reverse in further steps of carcinogenesis. Furthermore, progression of adenoma to advanced adenoma may be in general very likely (41.81–44.95%). Similarly, progression from early to advanced cancer is highly probable (50.54–73.73%). On the other hand, TR for progression from advanced adenoma to CRC is less than 10% (7.67–9.02%). This “dent” in dynamics of progression from benign to cancerous lesion probably represents biological and genetic changes that must occur for malignant transformation.

Lastly, we have calculated prevalence ratios. As shown in Figure 3, we observed more than expected adenomas and advanced adenomas in obese individuals, but less than expected early and advanced cancers. These results further contribute to a notion that a phenomenon referred to as obesity paradox exists in colorectal cancer. It is usually illustrated as decreased mortality in obese CRC patients, contrary to expected increase. While initially such findings were explained majorly by methodological biases, recently some authors point that these rationalizations may not be sufficient [21,22,23,24,25]. This can lead to a supposition that actual biological phenomena are being observed. Our results further contribute as a proof towards that supposition.

Present study is a sequelae of a previously published analysis performed by our team [9], where we explored distribution of CRC stages stratified by BMI categories in asymptomatic patients diagnosed within PCSP. We have found that obese men are more likely to be diagnosed with early cancer. In the discussion we have hypothesized that this may be either due to increase in adenoma formation or decrease of CRC progression, and thus have designed the present study to test these hypotheses. Based on results of the present study both of these hypotheses are likely to be true, i.e., excess of early cancers observed in obese individuals is both due to enhanced adenoma formation and slower CRC progression. Possible explanation of slower progression include unfavorable microenvironments in the fatty liver and lymph nodes and insulin resistance [26,27,28]. Therefore, present analysis provides new epidemiological rationale for basic research focused on exploring biological, genetic and metabolic differences between CRCs in obese and non-obese individuals. Such studies may both confirm and provide explanations for present findings. 

Main strengths of our study include substantial study group drawn from a prospectively maintained PCSP database. As previously mentioned [9], it enables analysis of asymptomatic individuals and gathering pre-diagnosis BMI, which in effect is the least influenced by the course of the disease. Therefore, presented findings should be less susceptible to paradoxical explanations. Further strengths include novel methodology that was used to illustrate the dynamics of adenoma-carcinoma sequence of CRC using confounder-adjusted data.

There are some limitations that need to be considered. As this is a screening-based study which facilitates questionnaires filled by screenees, it is susceptible to biases such as healthy screenee bias and response bias [29]. The latter especially needs to be considered as BMI in PCSP is a self-reported data and therefore may be underreported or omitted by screenees. This is of importance, as obese individuals are more likely to not report BMI when compared to the normal BMI individuals [30]. Moreover, use of BMI is a limitation itself and its use in the definition of obesity may be a major cause of the reported obesity paradox. BMI represents the sum of fat-mass index (FMI) and fat-free mass index (FFMI). The latter accounts for skeletal muscle mass, bone, and organs, while FMI is composed of peripheral and visceral adipose tissues. All these components of BMI may have different roles in contributing to progression of adenoma to cancer, and changes in BMI are not related to a proportional and linear modification of body compartments. Further limitation is a relatively small study group in advanced CRC subgroup, what has led to a possibly underpowered analysis yielding non-significant results in this category. Additionally, present study is based on data from the opportunistic arm of Polish Colonoscopy Screening Program. This may bias the data, as some of the patients, although claiming opposite, may have actually been symptomatic at the time of screening. This is much less likely in an invitation-based screening.

Finally, the most important consideration is the cross-sectional design. It provides a unique opportunity to observe the prevalence of specific colorectal lesions and analyze the relationships between prevalence among subgroups, such as BMI categories. However, since natural history is not actually observed in one individual, presented findings can be only interpreted as estimates, not actual values. This is of importance especially for (estimate) transition rates presented in the current study. These values should not be treated as actual risk of progression between steps of carcinogenesis, but rather as a proxy to describe the dynamic of the process and difference between BMI categories.

## 5. Conclusions

In conclusion our study provides new evidence that obesity paradox exists for colorectal cancer. It seems that obese individuals are more likely to develop adenomas, advanced adenomas and CRC but might be less likely to progress to advanced CRC. Furthermore, this study provides a rationale for basic research focusing on biological explanation of presented phenomena. However, due to study limitations listed above our results should be interpreted with caution. Future confirmatory studies are warranted.

## Figures and Tables

**Figure 1 cancers-14-00062-f001:**
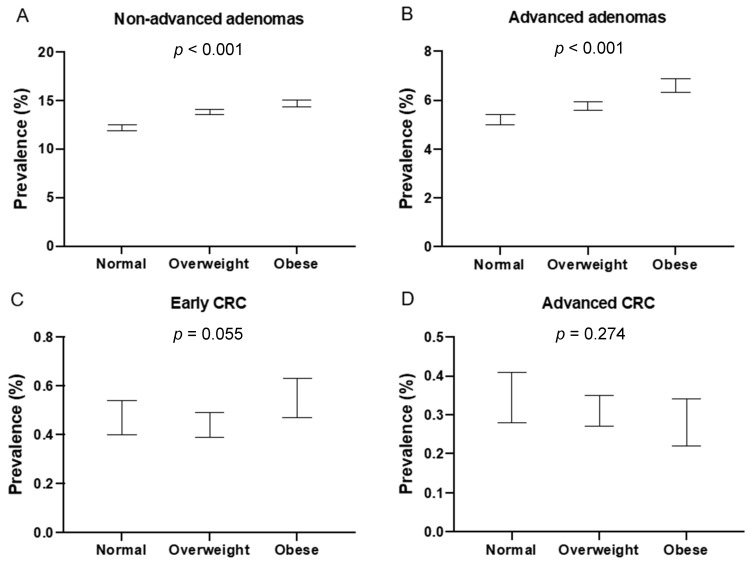
Comparison of adjusted prevalence: (**A**): adjusted prevalence of non-advanced adenoma stratified by BMI categories; (**B**): adjusted prevalence of advanced adenoma stratified by BMI categories; (**C**): adjusted prevalence of early CRC stratified by BMI categories; (**D**) adjusted prevalence of advanced CRC stratified by BMI categories. Error bars represent 95% confidence interval.

**Figure 2 cancers-14-00062-f002:**
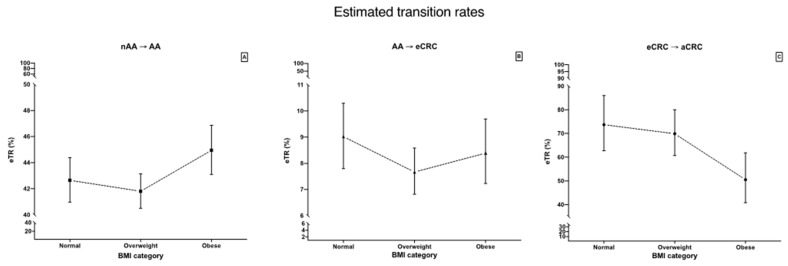
Comparison of eTRs stratified by BMI: (**A**)—non-advanced adenoma to advanced adenoma eTR (eTR_nAA__→__AA_); (**B**)—advanced adenoma to early CRC (eTR_AA__→__eCRC_); (**C**)—early CRC to advanced CRC eTR_eCRC__→__aCRC_). Error bars represent 95% confidence interval. BMI is a categorical variable—dotted lines are for visual purposes only. Note that y-axis is zoomed in.

**Figure 3 cancers-14-00062-f003:**
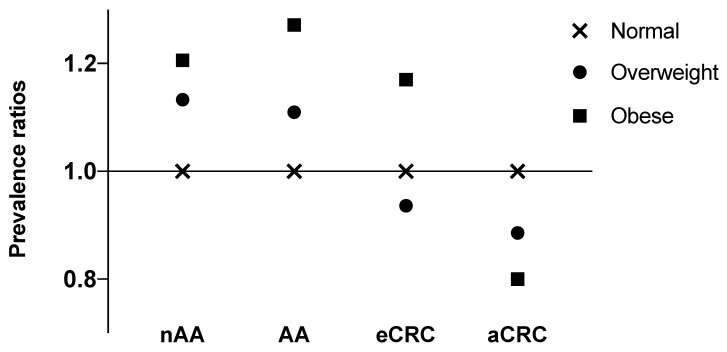
Prevalence ratios for BMI category and types of lesions. nAA—non-advanced adenoma, AA-advanced adenoma, eCRC—early CRC, aCRC—advanced CRC.

**Table 1 cancers-14-00062-t001:** Baseline characteristics of the studied population.

BMI Category	Normal *n* = 46,294	Overweight *n* = 67,528	Obese *n* = 33,563	ALL *n* = 147,385
Age mean (SD)	55.35 (5.52)	56.39 (5.20)	56.84 (5.01)	56.17 (5.20)
Male gender *n* (%)	12,702 (27.44%)	30,286 (44.85%)	13,990 (41.68%)	56,978 (38.66%)
Family history of CRC *n* (%)	9667 (20.88%)	12,112 (17.94%)	5562 (16.57%)	27,341 (18.55%)
Diabetes *n* (%)	757 (1.64%)	2613 (3.87%)	3424 (10.20%)	6794 (4.61%)
Never smokers *n* (%) *	26,817 (57.93%)	37,650 (55.75%)	17,969 (53.54%)	82,436 (55.93%)
PPI use *n* (%)	6031 (13.03%)	8769 (12.99%)	4532 (13.5%)	19,332 (13.12%)
Aspirin use *n* (%)	4183 (9.04%)	9044 (13.39%)	6461 (19.25%)	19,688 (13.36%)
HRT use *n* (%)	7790 (23.19%)	8094 (21.73%)	3323 (16.98%)	19,207 (21.25%)

* remaining subset of patients are either current smokers or ever smokers. PPI—proton pump inhibitors, HRT—hormone replacement therapy

**Table 2 cancers-14-00062-t002:** Crude and adjusted prevalence of colorectal lesions.

	*n* Diagnosed	Crude Prevalence	Adjusted Prevalence	95% CI	*p*-Value *
nAA	19,974				<0.001
Normal	5304	11.46%	12.19%	11.86–12.52	
Overweight	9600	14.22%	13.81%	13.55–14.07	
Obese	5070	15.11%	14.70%	14.31–15.08	
AA	8594				<0.001
Normal	2295	4.96%	5.20%	4.98–5.42	
Overweight	4029	5.97%	5.77%	5.60–5.95	
Obese	2270	6.76%	6.61%	6.33–6.88	
eCRC **	709				0.055
Normal	199	0.43%	0.47%	0.40–0.54	
Overweight	314	0.47%	0.44%	0.39–0.49	
Obese	196	0.59%	0.55%	0.47–0.63	
aCRC **	445				0.274
Normal	134	0.29%	0.35%	0.28–0.41	
Overweight	213	0.32%	0.31%	0.27–0.35	
Obese	98	0.29%	0.28%	0.22–0.34	

* all categories; ** Unknown stage for 84, 124 and 75 patients from normal, overweight and obese, respectively (not included in the analysis).

## Data Availability

Data are not publicly available.

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
