# Peer review of "Adenoma to Colorectal Cancer Estimated Transition Rates Stratified by BMI Categories—A Cross-Sectional Analysis of Asymptomatic Individuals from Screening Colonoscopy Program"

_cancers, 2021, doi:10.3390/cancers14010062_

Round 1
Reviewer 1 Report
Authors performed analyses on 147 385 (!) individuals based on data from Polish Colonoscopy Screening Program. They clearly showed that obese patients are more likely to develop advanced adenomas and early colorectal cancer (CRC), but less likely to progress to advanced CRC. These findings have important clinical consequences. Notably, this is probably the first study to show that estimated dynamics of CRC progression throughout multistep carcinogenesis process differ between normal, and obese individuals.
The paper is well written, harbouring important clinical message, and suitable of publication in its present form.
MINOR: correction needed Abstract Row 37: “eTR for early CRC do (to) advanced CRC…”
Author Response
Thank You for this review.
The correction has now been made.
Reviewer 2 Report
My personal comments are:
Abstract is considered from readers the "business card" of the manuscript. As my concern, I found it a bit confused and I would suggest to improve it. Moreover, other little considerations are:
Line 17: bold is not necessary;
Line 18: authors refer to BMI; it should be added Body Mass Index.
Line 37: “do” should be changed with “to”.
Background: check english language.
Line 46: Remove "colorectal cancer" before CRC; the acronym is sufficient
Line 72: bold is not necessary.
Line 75: “Remove Polish Colonoscopy Screening Program” before PSCP.
Materials and Methods:
This section is well done.
Body mass index can be removed in line 107; BMI is ok.
Results:
The authors report that they have drawn a total of 163129 individuals from the database and that the 9.7% of them have been exluded for lack of informations. I think that percentage has to be recalculated.
Line 175: eTR acronym is sufficient.
Discussion:
Check English grammar
Author Response
Thank You for this review.
- "the aim" in line 17 is no longer in bold
- line 18 - BMI is now expanded
- line 37 - this is now corrected
- line 46 - colorectal cancer has been deleted, "CRCs" is now left
- line 72 - no longe in bold
- line 75 - removed
- line 107 - done
- results, recalculation - how do You mean?
(593+6129+1134+7888)/163129 = 0,09651257594 ~ 9,7% - line 175 - provided it was meant to be a comment on line 184 it is now corrected.
Whole manuscript has now been re-read by two authors and english grammar was improved.
Again, Thank You for this revision.
Reviewer 3 Report
The authors present the first study to show that estimated dynamics of CRC progression throughout carcinogenesis steps differ between normal, overweight and obese individuals, on a substantial group of asymptomatic individuals who participated in the national opportunistic colonoscopy screening program for CRC in Poland. This elegant and well-done study, illustrating the dynamics of adenoma-carcinoma sequence of CRC using confounder-adjusted data, provides new evidence that obesity paradox exists for colorectal cancer.
I have two considerations:
- A major cause of this obesity paradox may be the use of BMI in the definition of obesity. BMI represents the sum of fat-mass index (FMI) and fat-free mass index (FFMI). The latter accounts for skeletal muscle mass, bone, and organs, while FMI is composed of peripheral and visceral adipose tissues. All these components of BMI may have different roles in contributing to progression of adenoma to cancer, and changes in BMI are not related to a proportional and linear modification of body compartments. I suggest including a comment about it.
- In this study, eTR for early CRC to advanced CRC is lower for obese individuals in comparison to both normal and overweight individuals. What was CRC stage in individuals in the advanced CRC subgroup? Were there patients with advanced disease who accessed colonoscopy screening while asymptomatic?
Author Response
Thank You for this review.
1/ This comment has been added in discussion
2/
- All patients in the study are asymptompatic as per: title, line 76, line 93
- Advanced CRC is defined as CRC in clinical stage 3 and clinical stage 4 according to AJCC. Line 108-109
So yes - there were patients that were asymptompatic, accessed screening colonoscopy and were diagnosed with stage 3/4 CRC.